# Estimating IDF Curves Consistently over Durations with Spatial Covariates

**Jana Ulrich** [1,*], **Oscar E. Jurado** [1] , **Madlen Peter** [1], **Marc Scheibel** [2] **and Henning W. Rust** [1]

[1]  Institute of Meteorology, Freie Universität Berlin, Carl-Heinrich-Becker-Weg 6-12, 12165 Berlin, Germany; jurado@zedat.fu-berlin.de (O.E.J.); madlen.peter@met.fu-berlin.de (M.P.); henning.rust@fu-berlin.de (H.W.R.)

[2]  Wupperverband, Untere Lichtenplatzer Str. 100, 42289 Wuppertal, Germany; schei@wupperverband.de

[*]  Correspondence: jana.ulrich@met.fu-berlin.de

**Abstract:** Given that long time series for temporally highly resolved precipitation observations are rarely available, it is necessary to pool information to obtain reliable estimates of the distribution of extreme precipitation, especially for short durations. In this study, we use a duration-dependent generalized extreme value distribution (d-GEV) with orthogonal polynomials of longitude and latitude as spatial covariates, allowing us to pool information between durations and stations. We determine the polynomial orders with step-wise forward regression and cross-validated likelihood as a model selection criterion. The Wupper River catchment in the West of Germany serves as a case study area. It allows us to estimate return level maps for arbitrary durations, as well as intensity-duration-frequency curves at any location—also ungauged—in the research area. The main focus of the study is evaluating the model performance in detail using the Quantile Skill Index, a measure derived from the popular Quantile Skill Score. We find that the d-GEV with spatial covariates is an improvement for the modeling of rare events. However, the model shows limitations concerning the modeling of short durations $d \leq 30$ min. For ungauged sites, the model performs on average as good as a generalized extreme value distribution with parameters estimated individually at the gauged stations with observation time series of 30–35 years available.

**Keywords:** extreme value statistics; extreme precipitation; subdaily precipitation extremes; intensity-duration-frequency curve; duration-dependent GEV; vector generalized linear model; spatial covariates

## 1. Introduction

Extreme precipitation events are often associated with hazards such as flooding and the resulting damage. In Germany, many destructive floods have occurred in recent decades, i.e., the Elbe floods in 2002 and 2013. Weather conditions favoring the occurrence of heavy rainfall events are likely to increase with global warming [1] in Germany as well as in many places worldwide [2]. Therefore, it becomes even more important to adequately estimate occurrence probabilities of precipitation amounts and intensities, as this information is needed for the design of water management systems. These range from urban drainage systems to river and creek design and retention basins. Therefore different stakeholders need information on the occurrence of extreme precipitation for different durations. Consequently, it is necessary to understand the relationship between precipitation intensity, duration, and exceedance-probability.

For a single location, this relationship can be represented graphically in intensity-duration-frequency (IDF) curves, a commonly used tool for the design of hydrological structures [3]. However, there is no uniform procedure for estimating IDF curves, and different

countries have different regulations for which method to use. In Germany, IDF curves for the entire state region are currently provided by KOSTRA-DWD [4], a project of the German Meteorological Service. The KOSTRA-DWD IDF curves are the results of a multi-step procedure and a set of different strategies for different ranges of durations [5]. In the USA, the National Weather Service provides estimates of precipitation frequency via an online portal [6]. These estimates are based on a regional frequency analysis [7,8]. The model used by the Swiss Weather Service is based on a seasonal Bayesian approach [9]. The results are also made available online [10]. Recent developments also suggest a wide range of methods, such as the use of radar data [11], cluster analysis to group stations [12] or support-vector machines to estimate extreme events based on reanalysis data [13].

In statistics, the definition of extreme events is based on their rare occurrence. Their statistical analysis is therefore based on small samples and it is necessary to use those efficiently in order to extrapolate from observed to unobserved levels of intensity. Extreme value theory provides several approaches to this problem (for an introduction, see Coles [14]). In geosciences, the block-maxima approach is popular. This approach is based on modeling the probability distribution of block-maxima (e.g., monthly or annual maxima) with a generalized extreme value (GEV) distribution. The longer the time series are, the more reliable the estimates are. Even if relatively long time series of 50 years or more exist at many places in Germany for daily precipitation sums, similarly long time series for observations at shorter durations are still an exception, since recording at such high frequencies is based on relatively new technology. Therefore, pooling existing information across duration can be beneficial.

In this study, we model both spatial variations of the probability distribution as well as its dependence on the accumulation duration in a consistent way. This approach allows us to include data of several gauge stations and a range of durations simultaneously in our estimation and hence makes efficient use of the available data. Instead of modeling the probability distribution individually for different precipitation durations, Koutsoyiannis et al. [15] proposed a duration-dependent distribution based on empirical dependencies of distribution parameters on duration. This approach provides the advantages of parameter parsimony and the direct availability of estimates for all durations within the interval considered. This was already employed in previous studies [16–18]. Similar to the studies of Lehmann et al. [17], Stephenson et al. [19], Blanchet et al. [20], who used a single model for a wide range of durations, we model a duration range spanning from one minute to five days. Thereby, we aim to transfer knowledge from the long durations, for which long time series exist, to the short durations.

Extending the model to include spatial variations not only provides the opportunity to estimate the IDF relationship for several locations simultaneously, but we expect that pooling information from several stations will reduce the uncertainties of parameter estimation, especially for stations with short observation time series. Many different statistical methods are used to model the spatial variation of the IDF relationship. The most straightforward way would be the spatial interpolation of the estimated distribution parameters, as done in [20]. A commonly used approach is regional frequency analysis, which combines data from stations with similar characteristics [8]. In contrast, spatial variations can be modeled in a single step, using Bayesian Hierarchical Models (BHM) [17,21,22], Vector Generalized Linear Models (VGLM) [23,24], or Vector Generalized Additive Models (VGAM) [25], which simplifies the estimation of uncertainties. The BHM's provide the uncertainty estimates directly, while for VGLM's and VGAM's, they can be obtained using, for example, the bootstrap method [26]. Fischer et al. [23] used a GEV to model daily precipitation sums and showed that the inclusion of Legendre polynomials for longitude, latitude, and altitude as covariates in location, scale and shape parameter contributed to a considerable improvement of the model compared to station-wise modeling.

Here we use the idea of Koutsoyiannis et al. [15] in the framework of VGLMs to combine the modeling of multiple durations and spatial variations by integrating orthogonal polynomials of longitude and latitude as covariates to describe the spatial variability of the parameters of a duration-dependent GEV (d-GEV).

We expect that this will allow us to provide estimates for all durations within the range that is used for parameter estimation and, to a certain extend, also to extrapolate beyond. Furthermore, we obtain

IDF relations at ungauged sites and improve the estimates for locations and durations with existing but short time series. To verify these assumptions, we test the approach in the study area of the Wupper catchment in the West of Germany and use the Quantile Skill Score [27] in a cross-validation setting [28] to evaluate the model performance for a range of return periods and individually resolved for all durations. We focus on two research questions:

1. Under which conditions is the spatial d-GEV approach an improvement compared to the separate application of the GEV for each duration and station?
2. Does the spatial d-GEV approach provide reliable estimates at ungauged sites?

In Section 2, we describe the data on which the study is based and the methods used for modeling, i.e., parameter estimation, model selection, estimation of confidence intervals and verification. The verification results are presented in Section 3.1. Return level maps and IDF curves are provided in Section 3.2. The results are discussed in Section 4, the last section summarizes methods, results and conclusions.

## 2. Methods

We integrate spatial covariates for the parameters of a duration-dependent GEV (d-GEV) to model extreme precipitation, both in space and over a range of durations. As a case study, we use data form the area of the Wupper River catchment. The covariates for the d-GEV parameters are selected through step wise forward regression and the model results are then verified using the Quantile Skill Score. Finally, confidence intervals for the IDF curves can be obtained using the bootstrap method. This section presents the data and describes the methods used throughout this study.

### 2.1. Data

We carry out a case study in an area surrounding the catchment of the Wupper River in North Rhine-Westphalia in western Germany. For this purpose we use precipitation measurements from 92 gauge stations, shown in Figure 1. The Wupper River is a right tributary of the Rhine with a length of 116 km, whereby the Wupper catchment, represented by a black line, has a moderate area of 813 km$^2$. As the area extends from the Cologne-Bonn lowlands in the west to the Bergisches Land in the east, different altitudes are well represented by the stations and a great variability in topographic shapes is covered.

The used gauge stations are operated by two different institutions: the German Meteorological Service (ftp://ftp-cdc.dwd.de/climate_environment/CDC/observations_germany/climate) (DWD) and the Wupperverband (https://www.wupperverband.de/) (WV). The station properties are summarized in Table 1. Notably, the measuring intervals at the respective stations differ.

Observations were accumulated to multiples of their original measuring interval, resulting in time series for 15 different durations: $1, 4, 8, 16, 32$ min, $1, 2, 3, 8, 16$ h and $1, 2, 3, 4, 5$ days. For each station the annual maxima of these respective time series were considered. Whereby, years with more than a total of 20 days of missing values were discarded. For gauge stations that are within 250 m of each other and do not vary more than 10 m in height, the measurements were grouped, to avoid very high correlations of the annual maxima. For stations that are grouped together, only those available values per year resulting from the higher measurement frequency were taken into account. Hence our data set contains a total of 24,304 annual precipitation maxima for all stations and durations combined. We provide the annual maxima as a data set online as supplementary material.

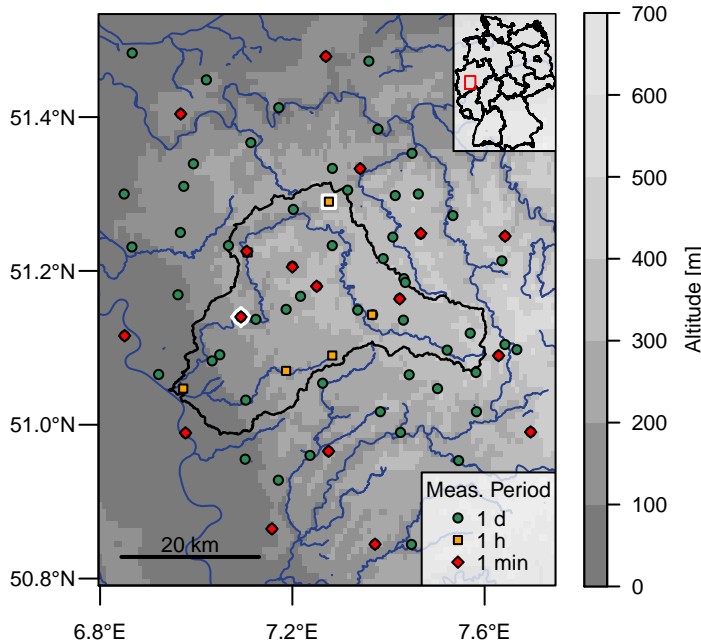

**Figure 1.** Study area containing 92 gauge stations with different measurement periods. The black line borders the Wupper catchment. Gauges marked white are those used as example stations (Schwelm (square) and Solingen-Hohenscheid (diamond)). The altitude is coded along a grey scale and stems from http://www.diva-gis.org/gdata, river shapes come from https://www.openstreetmap.org.

**Table 1.** Properties of precipitation gauge stations.

| Provider | Number of Stations | Measuring Interval | Device | Length of Time Series |
|----------|--------------------|--------------------|--------|-----------------------|
| DWD | 69 | 1 day | Hellmann | 9–121 years |
| DWD | 17 | 1 min | Pluvio | 5–14 years |
| WV | 6 | 1 h | Pluvio | 38 years |

*2.2. d-GEV as a Model for Annual Maxima for Different Durations*

Extreme value theory (EVT) provides methods for the statistical description of the tail of probability distributions and thus allows to estimate probabilities of very rare or even not yet observed events. The following descriptions are based on the introduction by Coles [14]. The basis of EVT is the Fischer-Tippett-Gnedenko Theorem, which essentially states that under certain assumptions the probability distribution of block maxima can be approximated by the generalized extreme value (GEV) distribution. More precisely, for $n$ independent and identically distributed copies $X_i$ of a random variable $X$, we define the block-maximum as

$$M_n = \max\{X_1, ..., X_n\}. \tag{1}$$

If for block-size $n \rightarrow \infty$ the distribution of the properly normalized $M_n$ converges to a non-degenerate distribution, then for a finite but large $n$ the non-exceedance probability

$$\Pr\{M_n \leq z\} \approx G(z), \tag{2}$$

can be modeled with the generalized extreme value distribution (GEV)

$$G(z; \mu, \sigma, \xi) = \exp\left\{-\left[1 + \xi\left(\frac{z-\mu}{\sigma}\right)\right]^{-1/\xi}\right\}, \tag{3}$$

defined on $\{z : 1 + \xi(z - \mu)/\sigma > 0\}$, with location parameter $\infty < \mu < \infty$, scale parameter $\sigma > 0$ and shape parameter $-\infty < \xi < \infty$, $\xi \neq 0$. Therefore the GEV can be used to model the annual precipitation intensity maxima for a certain precipitation duration $d$, e.g., daily precipitation sums.

### 2.2.1. Station-Wise Model for a Range of Durations (d-GEV)

In order to describe the relationship between precipitation intensity, duration and frequency, (i.e., non-exceedance probability), it is necessary to model the precipitation intensity maxima over a range of durations. The classical approach consists of two sequential model steps [29,30]. The first step is to separately estimate an extreme value distribution, e.g., the GEV, for a certain number of durations. In the next step, certain selected quantiles of the individual distributions are fitted using an empirical function with two to three parameters, which describes the relationship between intensity and duration. A detailed summary of frequently used empirical functions can be found in [31]. Koutsoyiannis et al. [15] demonstrated that all these empirical models are special cases of the more general form with 4 parameters

$$q_p = \frac{\omega}{(d^\nu + \theta)^\eta}, \tag{4}$$

with intensity quantile $q_p$ corresponding to the non-exceedance probability $p$, duration $d$ and with the non-negative coefficients $\omega, \nu, \theta, \eta$, where $\nu\eta \geq 1$. They showed furthermore that the assumption $\nu = 1$ is a sufficiently good approximation, resulting in a three parameter IDF model.

Their proposal is to implement Equation (4) with $\nu = 1$ directly into the parameters of the used extreme value distribution, to estimate the IDF curves in a considerably more consistent approach. Thus, an extreme value distribution is obtained in one step that is valid for a whole range of durations. We follow the ideas of Koutsoyiannis et al. [15] for the dependence of the GEV (Equation (3)) parameters on duration:

$$\sigma(d) = \sigma_0(d + \theta)^{-\eta}, \tag{5}$$
$$\mu(d) = \tilde{\mu} \cdot \sigma(d), \tag{6}$$
$$\xi(d) = \text{const.} \tag{7}$$

The dependence of location and scale parameters on duration is described using duration offset $\theta \geq 0$ and duration exponent $0 < \eta \leq 1$. Furthermore, $\sigma_0 > 0$ can be interpreted as a scale offset, since it indicates the scale parameter of the GEV distribution at $d = 1 - \theta$. Re-parameterizing the location $\tilde{\mu} = \mu(d)/\sigma(d)$ and inserting relation Equation (5) into Equation (3) results in a duration-dependent generalized extreme value distribution (abbreviated as d-GEV)

$$G(z, d; \tilde{\mu}, \sigma_0, \xi, \theta, \eta) = \exp\left\{-\left[1 + \xi\left(\frac{z}{\sigma_0(d + \theta)^{-\eta}} - \tilde{\mu}\right)\right]^{-1/\xi}\right\}. \tag{8}$$

This model describes the distribution of precipitation block maxima for a whole range of durations, with only two additional parameters than the GEV for one single duration.

The distribution's quantiles $q_{d,p}$ for a duration $d$, corresponding to the non-exceedance probabilities $p$, are equal to intensity-duration-frequency (IDF) relationships

$$q_{d,p} = \frac{\tilde{\mu}\sigma_0}{(d + \theta)^\eta} + \frac{\sigma_0}{\xi(d + \theta)^\eta}\left[1 - \{-\log(p)\}^{-\xi}\right]. \tag{9}$$

Whereby the parameters $\theta$ and $\eta$, respectively, describe the curvature for short durations and the slope for longer durations of the resulting IDF curves shown in a double-logarithmic plot. Hence, we can use the d-GEV to model annual precipitation intensity maxima at a single station over a range of durations simultaneously. In the following we will call this approach station-wise d-GEV. Its advantage is the reduction of the number of parameters needed to be estimated. More precisely,

$3n_d + 3n_q$ parameters would be required to model $n_q$ IDF curves using Equation (4) with $\nu = 1$. This is because we would first have to estimate the three parameters of the GEV distribution (Equation (3)) for a number of durations $n_d$ and then the three parameters of Equation (4) for each quantile. In contrast, we only need to estimate five parameters to model the distribution for all durations as well as the dependency of any quantile on duration in one step using the d-GEV (Equation (8)).

Additionally, we consider a special case for time series with sampling interval $d \geq 1\,\text{h}$ (e.g., hourly or daily): As the curvature (departure of a straight line) of the IDF relation shown in a double-logarithmic plot is only visible for small durations $d < 1\,\text{h}$, we assume $\theta = 0$ (i.e., no curvature) for durations $d \geq 1\,\text{h}$. Consequently, for gauges with observations sampled at hourly or longer sampling interval, $\theta$ is not estimated but set to zero.

### 2.2.2. Adding Spatial Covariates

The station-wise d-GEV approach already enables interpolation and pooling of information across durations. We further extend this approach in the framework of vector generalized linear models (VGLM) [25], to additionally allow for interpolation and pooling of information between gauge stations. We therefore model the spatial variations of every d-GEV parameter $\phi \in \{\tilde{\mu}, \sigma_0, \xi, \theta, \eta\}$ using a generalized linear model (GLM) of the form

$$l^\phi(\phi) = \phi_0 + \sum_{i=1}^{I} \beta_i^\phi x_i, \tag{10}$$

with the parameter specific link function $l^\phi(\cdot)$, intercept $\phi_0$ and regression coefficients $\beta_i^\phi$ and the covariates $x_i$. We implemented a function for parameter estimation based on maximizing the likelihood for the d-GEV with spatial covariates. This function is available as package `IDF` for the `R` environment [32,33]. Typically the choice of a link function ensures parameters to be positive or within a predefined range. Here, we implemented intervals for the parameters directly into the optimizer and thus used the identity $l^\phi(\phi) = \phi$ as link function for all parameters.

Following Fischer et al. [23], we intended to use orthogonal polynomials of longitude, latitude and altitude as covariates for the d-GEV parameters to model the spatial variations. However, since the area of our investigation is small and longitude and altitude are highly correlated in this area, we only use orthogonal polynomials of longitude and latitude. These polynomials are produced using the function `poly` from the package `stats` in the `R` environment [33]. We also add interactions resulting from the products of the respective terms. This yields the following model for each d-GEV parameter

$$\phi = \phi_0 + \sum_{j=1}^{J} \beta_j^\phi P_j(\text{lon}) + \sum_{k=1}^{K} \gamma_k^\phi P_k(\text{lat}) + \sum_{j=1}^{J} \sum_{k=1}^{K} \delta_{j,k}^\phi P_j(\text{lon}) P_k(\text{lat}). \tag{11}$$

Considering the size of the study area, we expect that a maximum order of $J = K = 6$ is sufficient to model the spatial variations within. We maximize the likelihood to obtain estimates of the intercepts $\phi_0$ and regression coefficients $\beta_j^\phi, \gamma_k^\phi, \delta_{j,k}^\phi$.

Equation (11) models the spatial variation (with longitude and latitude) of d-GEV parameters; in general, it takes on different values for different stations $s$ as longitude and latitude vary with station location. This results in a different d-GEV distribution at every station $s$ (and in between stations at arbitrary locations (lon,lat)).

Assuming independent observations (even across durations) leads to a factorization of the likelihood for the parameters given the observations $\mathbf{Z}$

$$\mathcal{L}(\tilde{\boldsymbol{\mu}}, \boldsymbol{\sigma_0}, \boldsymbol{\xi}, \boldsymbol{\theta}, \boldsymbol{\eta} \mid \mathbf{Z}) = \prod_{s \in S} \prod_{d \in D} \prod_{n \in N} g(z_{s,d,n}, d \mid \tilde{\mu}_s, \sigma_{0_s}, \xi_s, \theta_s, \eta_s), \tag{12}$$

where $g$ is the probability density function of the d-GEV, $Z$ is a vector containing annual maxima for different stations $s$, durations $d$ and years $n$ and $\tilde{\mu}$, $\sigma_0$, $\xi$, $\theta$ and $\eta$ are vectors containing the d-GEV parameters at each station. Strictly speaking, Equation (12) is only valid under the assumption that the observations in $Z$ are independent. We are aware that there is some asymptotical dependence between the annual maxima at nearby stations (for an overview on the topic, see [21] ), as well as between the annual maxima for different durations [34,35]. However, Jurado et al. [35] showed that the use of a model for IDF curves that accounts for asymptotical dependence between durations had a limited improvement on the performance of predicted return levels, with the increased model complexity of a max-stable process. Davison et al. [21] also suggests that spatial dependencies can be neglected for the estimation of point-wise return levels. Throughout our study, we will assume independence between the annual maxima of different durations and neighboring stations.

To obtain a parsimonious model, we use a selection procedure to determine which of the regression coefficients in Equation (11) are actually needed.

### 2.3. Model Selection

Choosing $J = K = 6$ in Equation (11) as the maximum order of the orthogonal polynomials for modeling the five d-GEV parameters results in 48 terms in the predictor (covariates) and thus for the 5 parameters yielding $2^{5 \cdot 48}$ possible models. Hence, the model selection is a challenging and also crucial task. We apply a step-wise forward regression, where we iteratively add one covariate to each of the parameters in a predefined order. If the addition of the covariate to the parameter model results in a better score, the model is augmented with this covariate. Since the d-GEV parameters are not independent of each other, the order in which the covariates are added strongly influences the result of the model selection. We chose the sequence $\theta \rightarrow \eta \rightarrow \sigma_0 \rightarrow \tilde{\mu} \rightarrow \xi$, according to the order in which the parameters occur in Equations (5)–(7). However, this is just one of many possible options that could be considered. A more efficient strategy for model selection based on boosting [36] is currently under investigation for this use case.

We use the cross-validated likelihood as a criterion for model selection. For this purpose, we carry out a $k$-fold cross-validation with a small number of folds $k = 2$, as suggested by Arlot and Celisse [37]. To ensure independence of the cross-validation sets and equal distribution of data from different stations and durations, the cross-validation subsets are drawn from every other year. The forward selection is stopped when the cross-validated likelihood stops increasing.

This approach results in a model with 24 coefficients in total. The covariates selected for each of the d-GEV parameters, are presented in Figure 2 along with the order of their selection. The parameters differ greatly in the number of covariates. From the large number of covariates in $\eta$ we conclude that this parameter varies particularly strongly in space.

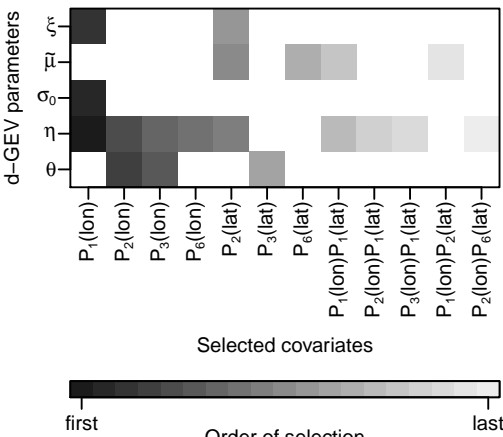

**Figure 2.** Final model selection result. For each d-GEV parameter, the added covariates are shown as colored boxes, according to the order of their selection.

*2.4. Verification*

Since estimates of the maxima distributions' tail (upper quantiles) are particularly important in this case, we evaluate the performance of the d-GEV model through cross-validation using the Quantile Score (QS) [27]. This score allows a detailed analysis for individual non-exceedance probabilities. The QS associated with exceedance probability $p$ is the weighted mean difference between observations $o_n$ and the modeled quantile $q_p$

$$\text{QS}(p) = \frac{1}{N} \sum_{n=1}^{N} \rho_p(o_n - q_p),$$  (13)

where the check loss function $\rho_p$ is defined as

$$\rho_p(u) = \begin{cases} pu & , u \geq 0 \\ (p-1)u & , u < 0. \end{cases}$$  (14)

Therefore, $\text{QS} \geq 0$, is negatively oriented with optimal value at zero. Thus the model performance can be examined for any non-exeedance probability $p \in (0, 1)$. However, the result may be less reliable when only few data are available for the verification of high probabilities (upper quantiles).

We use two different verification strategies: (1) to assess the performance in detail, we calculate QS at each station $s$ for a certain duration $d$ and probability $p$

$$\text{QS}_{s,d}(p) = \frac{1}{|Y_{s,d}|} \sum_{y \in Y_{s,d}} \rho_p(o_{s,d,y} - q_{s,d}(p)),$$  (15)

where $|Y_{s,d}|$ is the cardinality of $Y_{s,d}$, the set of years the score is evaluated for; (2) for an overview of model performance in the study area, we average QS over stations

$$\overline{\text{QS}}_d(p) = \frac{1}{S_d} \sum_{s=1}^{S_d} \text{QS}_{s,d}(p),$$  (16)

where $S_d$ is the number of stations, for which a time series for duration $d$ exists. With a $k$-fold cross-validation experiment, we assess the model's out-of-sample performance using Equation (15) [28,38]. Hence, we partition the data into $k$ sets $Y_{s,d}$ containing each $n_{\text{val}} = |Y_{s,d}| = 3$ years of data. This results in a varying number of sets depending on the length of the time series. Successively, each set is used once for validation, the remaining for training. Therefore, the number of years used for training also depends on the length of the time series. For the station-wise model the training set consists of the remaining years of data at the station under investigation; for the spatial model the training set additionally contains all observations from all other stations.

Finally, we compare the cross-validated score of the model $QS^M$ with the cross-validated score of a reference model $QS^R$ using the Quantile Skill Score (QSS) [28]

$$\text{QSS}^M(p) = \frac{\text{QS}^M(p) - \text{QS}^R(p)}{0 - \text{QS}^R(p)} = 1 - \frac{\text{QS}^M(p)}{\text{QS}^R(p)},$$  (17)

with $\text{QS}_{s,d}(p)$ (cf. Equation (15)) to evaluate QSS at each station $s$ and $\overline{\text{QS}}_d(p)$ (cf. Equation (16)) to assess performance across the study area. $\text{QSS}^M(p) \in (-\infty, 1]$ is positively oriented and optimal at 1, representing the gain in performance ($\text{QS}^M(p) - \text{QS}^R(p)$) relative to the difference between a perfect model and the reference model ($0 - \text{QS}^R(p)$).

If, however, the model performs worse than the reference, QSS is negative and the interpretation less intuitive. We thus define a Quantile Skill Index (QSI) as a combination of the model's skill with respect to a reference $\text{QSS}^M(p)$ and the skill of the reference with respect to the model $\text{QSS}^R(p)$

$$\text{QSI}(p) = \begin{cases} \text{QSS}^M(p) & ,\text{QS}^M(p) \leq \text{QS}^R(p) \\ -\text{QSS}^R(p) & ,\text{QS}^M(p) > \text{QS}^R(p) \,. \end{cases} \tag{18}$$

Positive values still indicate a gain with respect to the reference while negative values now indicate a gain of the reference with respect to the model. Now, we have $\text{QSI} \in [-1, 1]$. In the following, we distinguish between the QSI calculated using the QS for each station $\text{QS}_{s,d}(p)$ and the QS averaged over stations $\overline{\text{QS}}_d(p)$ by referring to the later one as average QSI and using the notations $\text{QSI}_{s,d}(p)$ and $\overline{\text{QSI}}_d(p)$, respectively. As reference we always use the GEV for annual maxima at each station and duration separately.

### 2.5. Confidence Intervals

We obtain 95% confidence intervals for the estimated IDF relation at a given station *s*, by applying the ordinary non-parametric bootstrap percentile method [39]. In a simulation study (see Appendix B), we compare the coverage of the 95% confidence intervals for the estimated quantiles derived through both the bootstrap percentile and the delta method [14] under the assumption of no dependence between block maxima. We find that the coverage for the delta method intervals depends on the duration and probability of the quantile, for which the confidence intervals are estimated. In most cases, the coverage deviates strongly (upwards or downwards) from the nominal 95%. The bootstrapped confidence intervals, on the other hand, show a consistent behavior for different durations and probabilities with a reasonable coverage for large enough sample sizes (see Figure A1).

## 3. Results

We will first present the results of the verification. We use different variations of the methods presented in Section 2.4 to assess different aspects of the model performance. We then address the estimation of quantiles for spatial maps of return levels and IDF curves at selected stations; both with their associated uncertainty.

### 3.1. Model Performance

Addressing the two questions posed in Section 1, we investigate model performance using variations of the cross-validation experiment described in Section 2.4. An overview is given in Table A1.

### 3.1.1. Overall Performance

We calculate $\overline{\text{QSI}}_d(p)$ (cf. Equation (18)) using the mean quantile score over all stations $\overline{\text{QS}}_d(p)$ (Equation (16)) to asses the overall performance in the whole study area. We furthermore obtain $\overline{\text{QSI}}_d(p)$ for both, the station-wise d-GEV approach and the spatial d-GEV approach, to be able to compare them. For a detailed assessment, we use $\text{QSI}_d(p)$ for all durations used and for the range of non-exceedance probabilities $p \in \{0.5, 0.8, 0.9, 0.95, 0.98, 0.99, 0.995\}$ associated with return periods $T = \{2, 5, 10, 20, 50, 100, 200\}$ years. Results are presented in Figure 3 for both, the station-wise d-GEV (upper panel) and the spatial d-GEV (lower panel).

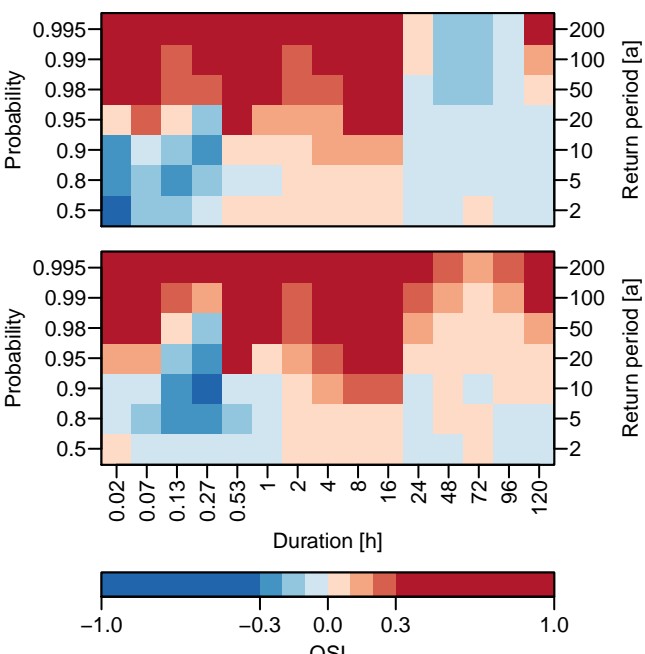

**Figure 3.** Average Quantile Skill Index $\overline{\mathrm{QSI}}_d(p)$ for different durations $d$ and probabilities $p$. Upper panel: station-wise d-GEV. Lower panel: spatial d-GEV. Positive values (red) indicate an improvement compared to the quantile estimates obtained by modeling each station and duration separately.

Both approaches show similar results. Overall, for probabilities $p \in \{0.5, 0.8, 0.9\}$ we see no or only a small improvement, while higher probabilities show improvements in model performance. This suggests that, since the station-wise and the spatial d-GEV approach make more use of the available data, they are both better for modeling rare events than the approach of modeling each duration at every station separately. Deviations from this pattern occur: First, strongly negative QSI values are observed for short durations $d < 1\,\mathrm{h}$ in a sequence of probabilities. Since this effect is apparent for both approaches, it is thought to be due to insufficient modeling of the curvature of the IDF curves in this range using the d-GEV. Second, for durations $d \geq 24\,\mathrm{h}$ the station-wise d-GEV approach leads to a loss of skill at almost all probabilities, and the spatial approach also shows a lower skill compared to the shorter durations. As mentioned in Section 2.1, the length of the time series varies strongly for different measurement intervals: where most stations with daily measurements have longer time series than stations with a shorter measurement intervals. We thus assume that the different behavior for longer durations $d \geq 24\,\mathrm{h}$ stems from the longer time series of daily observations used to train the model. In the subsequent section we examine this assumption in more detail.

### 3.1.2. Dependence on Time Series Length

A slightly modified verification method investigates the effect of time series length used for training. We divide each times series such that the series available for training at a particular station all have a fixed length $n_{\mathrm{train}}$. These are then used to train the model at this station. To this end, each station's data is split into blocks of years containing $n_{\mathrm{train}} + n_{\mathrm{val}}$ years each. Depending on the length of the time series for each station and duration, this will result in a varying number of blocks $b_{s,d}$. Again, $k$-fold cross-validation is used in each block to assess performance as previously described. The resulting $b_{s,d}$ cross validated scores at each station are then averaged to obtain $\mathrm{QS}_{s,d}(p)$. Thereafter, the analysis again follows as described in Section 2.4.

The resulting $\overline{\mathrm{QSI}}_d(p)$, obtained for different numbers of years $n_{\mathrm{train}}$ available to train the model at a certain station, is presented in the upper panel of Figure 4.

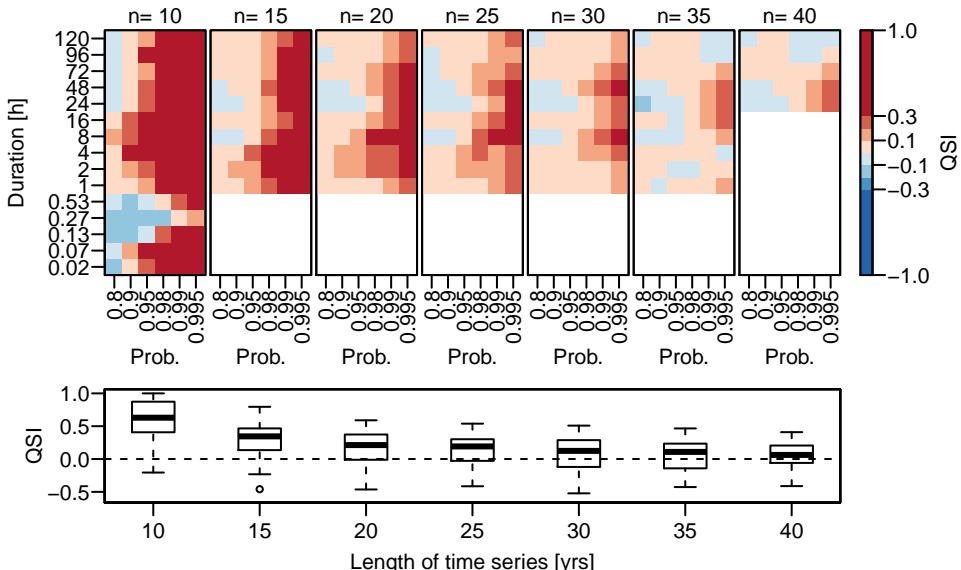

**Figure 4.** Dependence of the spatial d-GEV model performance on length of the training time series. Upper panel: Average Quantile Skill Index $\overline{\text{QSI}}_d(p)$ for different durations $d$ and probabilities $p$ as seen in Figure 3 but rotated. Different columns represent $\overline{\text{QSI}}_d(p)$ for different numbers of years in the training set $n_{\text{train}}$. Lower panel: Boxplot of the Quantile Skill Index $\text{QSI}_{s,d}(p)$ with probability $p = 0.99$ dependent on the length of the training time series.

Advantage and disadvantage of the spatial d-GEV is represented through colors red and blue, respectively; white indicates a situation with insufficient data to evaluate $\overline{\text{QSI}}_d(p)$. The two main features identified in Figure 3 recur: (i) an increase of $\overline{\text{QSI}}_d(p)$ with larger $p$ for all $n_{\text{train}}$ and (ii) a disadvantage of the spatial d-GEV for durations $d < 1$ h for a range of probabilities. Additionally, we observe a gradual decrease of the $\overline{\text{QSI}}_d(p)$ with $n_{\text{train}}$. Leading to the advantage of the spatial model in modeling rare events becoming smaller in each step until the average QSI fluctuates around zero. This indicates that there is approximately the same number of stations with gain as with loss in skill. This approach therefore provides us with the length of the time series of a station, up to which it is beneficial to use the spatial d-GEV model, instead of a separate GEV model for each time series. As an example, the $\text{QSI}_{s,d}(p)$ for $p = 0.99$ is shown in Figure 4 in the lower panel as a boxplot for various $n_{\text{train}}$. From this we can see that for $n_{\text{train}} = 35$ there are about as many stations with the spatial d-GEV being superior as stations where it is inferior to the reference model.

An alternative representation of the $\text{QSI}_{s,d}(p)$ is shown in Figure 5 for $p = 0.99$ and selected values for $d$ and $n_{\text{train}}$. Since the values for the QSI vary strongly, it is more difficult to detect the relationships between $\text{QSI}_{s,d}(p)$, $d$, $p$ and $n_t$ in this form of presentation. However, this way it is easier to observe that even with a large positive average QSI, negative QSI values can occur at individual stations. We further notice that for individual stations the sign of the QSI can be opposite for different durations. The average QSI decreases with increasing $n_{\text{train}}$, as more stations with negative QSI appear and the values at the stations with positive QSI become smaller.

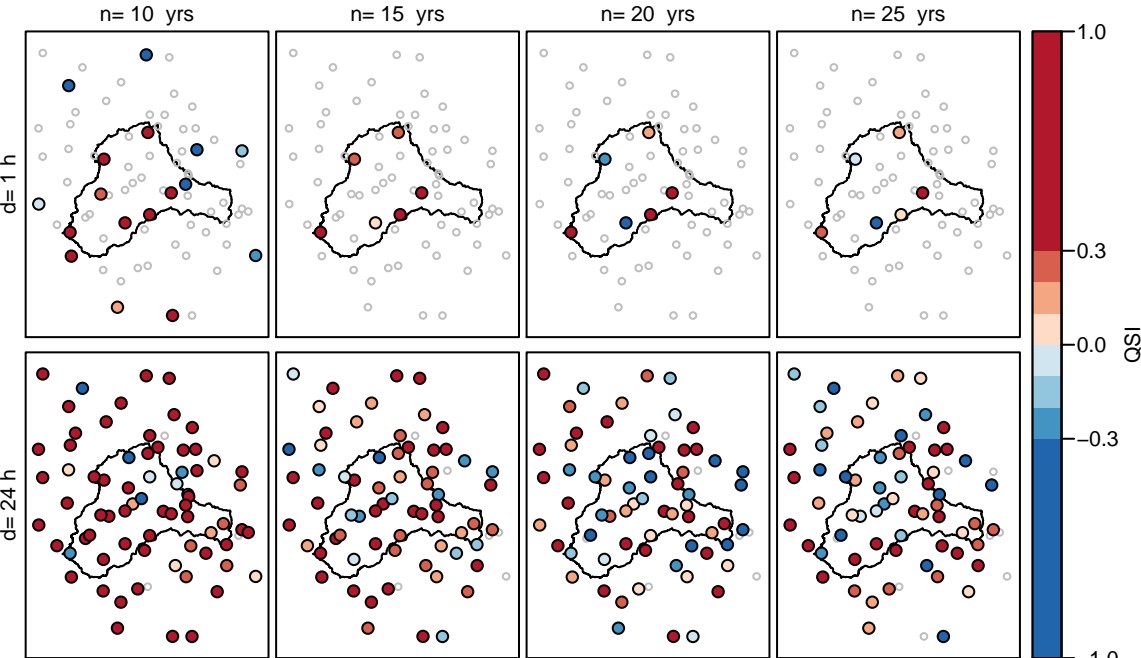

**Figure 5.** Quantile Skill Index $QSI_{s,d}(p)$ for the spatial d-GEV approach with probability $p = 0.99$ for durations $d = 1$ h (upper panel) and $d = 24$ h (lower panel) at all stations, for the number of training years $n_{train} = \{10, 15, 20, 25\}$ (different columns). Colored dots indicate superiority of the spatial d-GEV (red) or inferiority (blue), while gray circles show stations without an estimate.

### 3.1.3. Ungauged Sites

Model performance at ungauged sites can be assessed by disregarding all values at a certain station for training. Afterwards, the estimates at the location of the station excluded are compared to the actual observations. This procedure is carried out for each of the stations in the research area. However, we can only use this approach for the spatial model and not for the station-based reference model. Even though one might consider to compare the spatial d-GEV model to another spatial modeling approach, here we continue to use the GEV applied separately for each duration and station as reference model to keep the performance of the reference model as consistent as possible. We use the method described in the previous section to train the reference model with a fixed number of years at the station under investigation. Although the spatial d-GEV model is always trained on all available data except for the excluded station, we compare the model and reference with the same validation data set in each cross-validation step. Figure 6 shows the results for the average QSI with every column representing a different reference model associated to the number of years used for training, but the spatial d-GEV model in each column is trained on identical data.

This provides us with an approximate estimate of what is comparable to the performance of the spatial model at ungauged sites. Therefore we conclude that in our study area the estimates of higher quantiles $p \geq 0.98$ at ungauged sites is on average as good as the estimates based on the GEV for a station with measurements for 30–35 years. The comparison with Figure 4 reveals a strong similarity of the results. From this we conclude that adding $n_{train}$ years of data at one station only has a minor effect on the estimates of the spatial model on average. The differences between the columns in Figures 4 and 6 thus originate mainly from differences in the reference model.

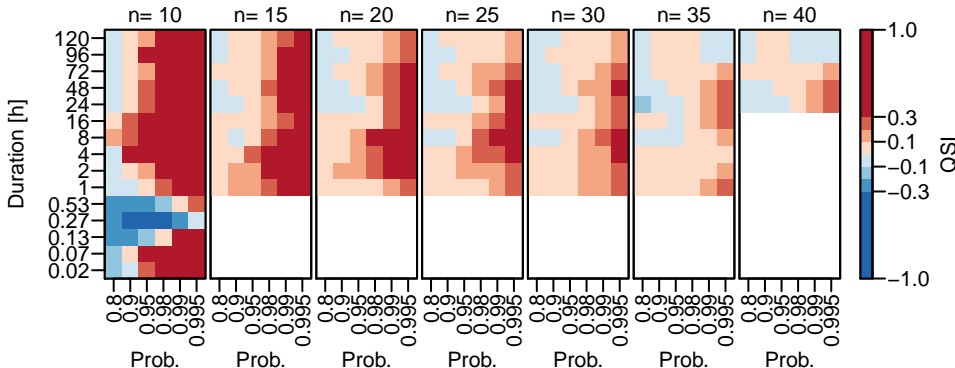

**Figure 6.** Average Quantile Skill Index $\overline{\mathrm{QSI}}_d(p)$ at ungauged sites for different durations $d$ and probabilities $p$, similar to Figure 4. For the spatial d-GEV the data at the respective stations are omitted for fitting, while the reference model uses a different number of years $n_{\mathrm{train}}$ in the training set (different columns.)

## 3.2. Quantile Estimation and Uncertainty

The spatial d-GEV allows for estimating the quantiles of the extreme value distribution at arbitrary locations in our research area for arbitrary durations. Hence we are able to provide return level maps for any desired duration. The 100-year and 20-year return level maps for the durations 5, 30 min and 2 h are presented in Figure 7.

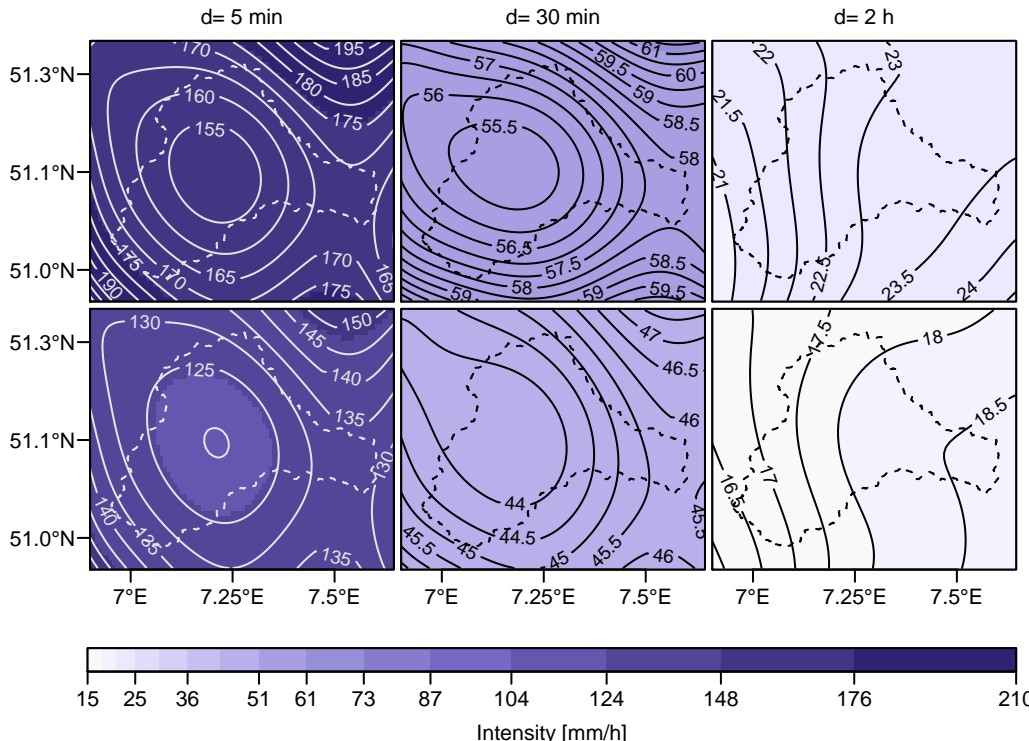

**Figure 7.** Point-wise return level maps for durations $d \in \{5, 30, 120\}$ min (different columns) and probabilities $p = 0.99$ (upper panel) and $p = 0.95$ (lower panel) corresponding to return periods of 100 years and 20 years, respectively. The colors are provided as a general reference between the plots.

With increasing duration, we observe a shift of the spatial patterns from a minimum in the center of the catchment to a west-east gradient ($d \geq 1$ h). In the Wupper catchment area there are three prominent weather conditions that occur together with precipitation: convective conditions from south-east and

south-west and advective conditions from north-west. The gradient in the intensity of extreme precipitation of 1 h or longer is plausible regarding the main direction of advective weather conditions and the increase in elevation towards the east in the study area. In contrast, the intensity patterns of the shorter events appear to be unrelated to the weather conditions mentioned above. The spatial variations in the precipitation intensity are larger for the short durations and the more rare events. Even though a number of stations placed in the surroundings of the catchment area were included into the modeling, high gradients can still be detected at the boundaries, resulting from extrapolation.

In addition to the return level maps, classical IDF curves can be obtained at any location in the study area. Exemplary, Figure 8 shows the IDF curves for the station Solingen-Hohenscheid.

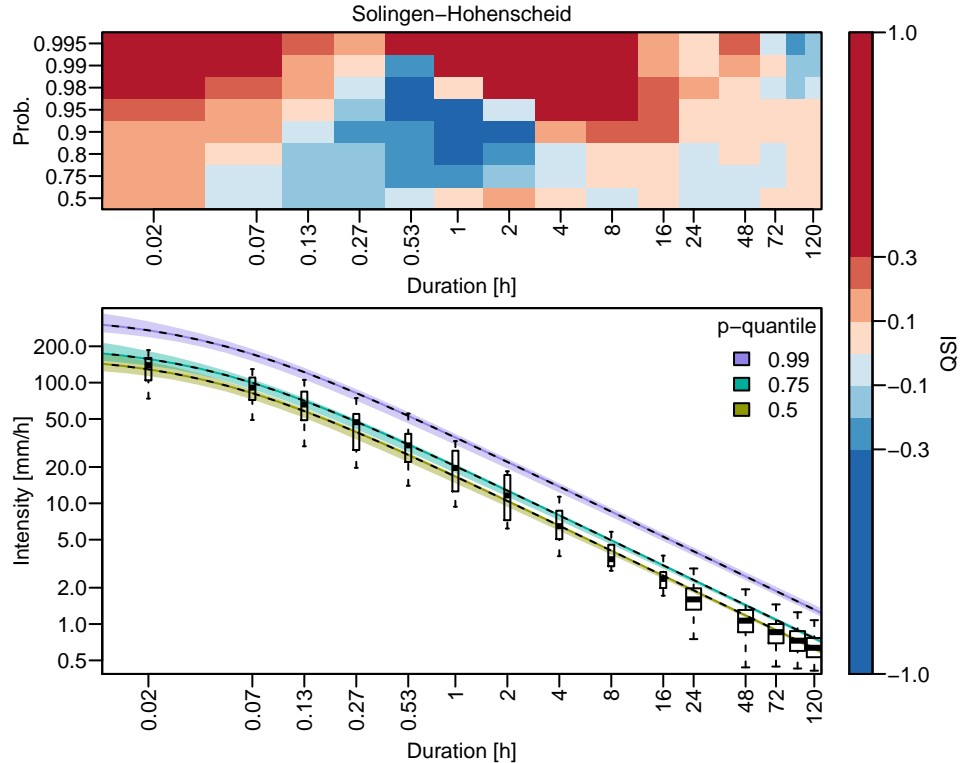

**Figure 8.** IDF-curve estimate for Solingen-Hohenscheid (marked with white square in Figure 1) (lower panel) obtained by using the spatial d-GEV (black dashed lines) and their 95% confidence intervals (colored areas). Observations are shown as boxplots, where the width of the box is proportional to the number of data points available at a certain duration. The upper panel shows the corresponding QSI values at that station following the presentation of Figure 3.

The curves represent $p \in \{0.5, 0.75, 0.99\}$, corresponding to return periods $T \in \{2, 4, 100\}$ years, together with their 95% confidence intervals. The observations—shown as boxplots—are well represented by the modeled IDF curves. To directly relate the estimated quantiles to the verification, the values of the QSI for this station are also shown in the upper panel. The QSI at this station exhibits similar structures as the average QSI. In the case of this station, the negative QSI values in the range 2 min to 4 h coincide with an underestimation of the respective quantiles by the spatial d-GEV. This suggests that the model lacks flexibility for this range of durations. However, the Quantile Score for these durations might not be very meaningful due to the limited amount of data.

The bootstrap confidence intervals are rather narrow across all durations. They represent the sampling uncertainty, assuming an adequate model has been selected for the data. The width of the intervals indicates that the uncertainty of the estimates is larger for shorter durations. Figure 9 compares the confidence intervals at small durations for the station-wise d-GEV and the spatial d-GEV at two example stations.

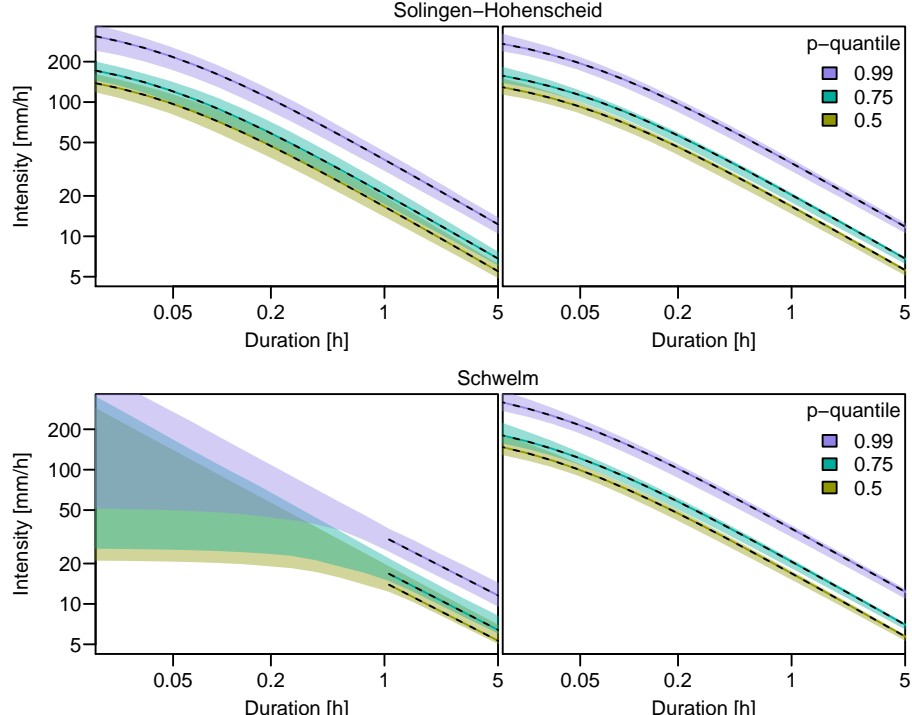

**Figure 9.** Bootstrapped 95% confidence intervals for IDF-estimates at the example stations Solingen-Hohenscheid and Schwelm (marked in Figure 1), using the station-wise d-GEV (left column) and the spatial d-GEV approach (right column).

At station Solingen-Hohenscheid data is measured at one minute intervals and at station Schwelm at hourly intervals. As a result of the increased data availability the confidence intervals based only on the uncertainties of the parameter estimation are smaller for the spatial model than for the station-wise model. The spatial d-GEV is additionally able to estimate the quantiles even for durations smaller than the measurement interval, based on the data of neighboring stations with relatively low uncertainties. This information is not available using the station-wise model.

## 4. Discussion

Since the QSI varies strongly between individual stations and individual durations, the assessment of the model performance and its presentation is challenging. However, the averaged QSI presented in Figure 3 allows for some conclusions. We find that the average QSI for the spatial d-GEV is strongly positive for upper quantiles (large non-exceedance probabilities $p \geq 0.98$). From this, we conclude that the spatial d-GEV approach is an improvement for modeling rare events since it benefits from the increased data availability at neighboring sites. However, for a range of smaller durations $d \leq 30$ min, the skill decreases for both the station-wise d-GEV and the spatial d-GEV model, compared to the reference which is based on an individual GEV for all stations and durations. This suggests that the d-GEV does not describe the variations in this range of small durations sufficiently well. Figure 8 presents the QSI together with the IDF estimates for gauge Solingen-Hohenscheid; this figure suggests that the negative QSI values in the range of 2 min to 4 h are related to an underestimation of the quantiles in this range. This supports the assumption that the d-GEV is not sufficiently flexible in this range and a more complex model might be necessary, as suggested in (e.g., [40]). Nevertheless, we only used time series with a maximum of 14 years to investigate the model performance for sub-hourly durations and thus cannot exclude the possibility that the effect may occur due to insufficient data.

Moreover, the average QSI is lower for durations $d \geq 24$ h. We believe this is due to a larger data availability for these durations and thus longer time series are available to train the reference model. This is supported by investigating the influence of time series length, where a fixed number of years

$n_\text{train}$ is used to train the model for each duration and at each station (cf. Figure 4). We observe a gradual decrease in the average QSI with the length of the training time series. We conclude that the advantage of the spatial d-GEV model over the station-wise GEV model is reduced for longer time series; The pooling of information becomes less important. We find that for a length of about 35 years, there are about as many stations with the spatial d-GEV being superior as stations where it is inferior to the reference model. This implies that in case of a single gauge with a long time series, the spatial d-GEV approach cannot outperform single site estimates for individual durations. However, due to the lack of data, we are unable to make any statements about the behavior of the estimates for $d < 1\,\text{h}$ with $n_\text{train}$. This information would be particularly helpful for these durations, as here often only short time series are available. Moreover, even with a strongly positive average QSI, negative values of the QSI for individual stations, and durations, occur. Yet, Fischer et al. [23] showed that even for long time series with more than 50 years of observations, a GEV model with spatial and seasonal covariates performed better than separate models for each month and station at almost all stations investigated. Therefore, the major improvement of the model by adding spatial and seasonal covariates is not directly applicable to the spatial d-GEV model.

However, a large advantage of the d-GEV model is its ability to interpolate between durations and stations at the level of distribution parameters and it is therefore possible to obtain estimates for durations and sites for which no measurements exists in a consistent way. This advantage has been disregarded in the verification process, where we used a separate model for individual durations and stations as a reference model. From the results presented in Figure 6, we infer that the model performance at ungauged sites is comparable to that of a separately applied GEV for an available time series of 30–35 years, at least for high quantiles $p \geq 0.98$. Therefore we conclude that the spatial d-GEV model provides reliable estimates at ungauged sites. However, the available time series for $d \leq 1\,\text{h}$ are again not long enough to investigate the model performance at ungauged sites for this range of small durations in this way.

## 5. Summary

In this study, we model annual precipitation maxima simultaneously in space and across durations. To this end, we integrate orthogonal polynomials of longitude and latitude as spatial covariates into the duration-dependent GEV proposed by Koutsoyiannis et al. [15]. This allows for a parameter parsimonious description compared to modeling of individual stations and durations, efficient use of existing data, and the pooling of information between stations and durations. We specifically model a wide range of durations from one minute to 5 days in order to investigate to what extent knowledge can be transferred from long observation time series and whether estimates for stations or durations with fewer observations benefit from this. We investigate this model in the Wupper catchment with the main focus on evaluating the model performance. Model validation is based on techniques from forecast verification: we use a variant of the Quantile Skill Score, the Quantile Skill Index (QSI). In the presentation used here, this score allows a detailed analysis of the model performance for different non-exceedance probabilities and durations, simultaneously. As a reference model that is not based on any empirical relationship between intensity and duration, the GEV is used to model precipitation maxima independently at individual stations and durations.

We find that using the spatial d-GEV improves the modeling of rare events of all durations, as it benefits from greater data availability. Accordingly, this model is advantageous for stations with short time series and does not necessarily improve the estimation if a longer time series is available. We also find that the d-GEV model is most likely not flexible enough to model the whole range of durations sufficiently and that a model with additional parameters (e.g., [40]) might be necessary. Therefore we recommend reducing the duration range in cases where the aim is exclusively the description of short durations $d \leq 1\,\text{h}$. Future studies will also explore the use of more flexible models to describe the whole range of durations. We expect that the estimation of further additional parameters for the duration dependence in these more flexible models will benefit from a spatial covariates setting.

Since this approach allows us to interpolate between stations and durations, spatial maps of return levels can be readily obtained for any duration, as well as IDF curves for any location in the research area. The bootstrap method provides 95% confidence intervals representing the sampling uncertainty. For the d-GEV with spatial covaiates, these uncertainties are smaller than for the station-wise d-GEV, since the spatial model can draw information from both neighboring sites and durations. Uncertainties from the model selection are not considered. For a reliable estimation of the uncertainties, Mélèse et al. [26] suggest a Bayesian Hierarchical Model. In the return level maps we observe that the spatial patterns change from a minimum in the center of the catchment for short durations to a west-east gradient for long durations. This is likely related to the main north-west direction of advective weather conditions in the study area and might also be linked to the orography.

In this work, we assume that there is no dependence between observations of different durations. This seems to be reasonably well justified as reported in another study which investigates the effect of including this dependence explicitly using a max-stable process for six stations in the same research area ([35] in this issue). We also assume that there is no dependence between observations at neighboring stations; this dependence could also be modeled using a max-stable process [19,21]. We use the assumption that the IDF relationship does not vary in time. However, we plan to account for the temporal variations of the IDF relationship by a straightforward extension of the spatial d-GEV model with further covariates in future studies. Nevertheless we could demonstrate that the approach presented here allows obtaining reasonable estimates of return levels for any arbitrary duration or location within the study domain, performing particularly well for rare events.

**Supplementary Materials:** The following are available online at http://www.mdpi.com/2073-4441/12/11/3119/s1 .

**Author Contributions:** Conceptualization, H.W.R. and J.U.; Data curation, J.U., O.E.J. and M.S.; Formal analysis, J.U.; Funding acquisition, H.W.R.; Methodology, J.U. and M.P.; Software, J.U., O.E.J. and M.P.; Visualization, J.U.; Supervision, H.W.R.; Writing—original draft, J.U.; Writing—review & editing, O.E.J., M.P., H.W.R. and M.S. All authors have read and agreed to the published version of the manuscript.

**Funding:** This study was developed within the framework of the research training program *NatRiskChange* funded by the Deutsche Forschungsgemeinschaft (DFG; GRK2043/1 and GRK2043/2) at Potsdam University and partially supported through grant CRC 1114 "Scaling Cascades in Complex Systems", Project A01 "Coupling a multiscale stochastic precipitation model to large scale flow". O.E.J. acknowledges support by the Mexican National Council for Science and Technology (CONACyT) and the German Academic Exchange Service (DAAD).

**Acknowledgments:** The authors would like to thank the Wupperverband as well as the Climate Data Center of the DWD, for providing and maintaining the precipitation time series. We would also like to acknowledge the constructive comments of the three anonymous reviewers and the academic editor that have contributed to improving the manuscript.

**Conflicts of Interest:** The authors declare no conflict of interest.

## Abbreviations

The following abbreviations are used in this manuscript:

| | |
|---|---|
| BHM | Bayesian Hierarchical Model |
| DWD | German Weather Service (Deutscher Wetterdienst) |
| EVT | Extreme Value Theory |
| GEV | Generalized Extreme Value distribution |
| d-GEV | duration-dependent GEV |
| IDF | Intensity-Duration-Frequency (curve) |
| QS | Quantile Score |
| QSS | Quantile Skill Score |
| QSI | Quantile Skill Index |
| VGLM | Vector generalized linear model |
| VGAM | Vector generalized additive model |

## Appendix A. Overview of Verification Variations

We perform the model verification with small variations in the cross-validation methods, to asses different aspects of the model performance:

1. the overall performance
2. the dependence of the model performance on the length of the time series used for training the model
3. the model performance at ungauged sites.

Table A1 provides an overview of the differences in the cross-validation sets used for training and validation in all three cases for the spatial d-GEV and the reference model, the GEV applied separately for each station and duration.

**Table A1.** Cross-validation sets used for training and validating the spatial d-GEV and the reference model (GEV). *N* represents the complete length of the time series and varies for different durations and stations, while $n_{\text{train}}$ is a fixed number of years for each time series.

| | | Overall Performance | | Dependence on Time Series Length | | Ungauged Sites | |
|---|---|---|---|---|---|---|---|
| | | Training | Validation | Training | Validation | Training | Validation |
| spatial d-GEV | station | $(N-3)$ years | 3 years | $n_{\text{train}}$ years | 3 years | - | 3 years |
| | remaining stations | all data | - | all data | - | all data | - |
| GEV | station | $(N-3)$ years | 3 years | $n_{\text{train}}$ years | 3 years | $n_{\text{train}}$ years | 3 years |
| | remaining stations | - | - | - | - | - | - |

## Appendix B. Coverage of Confidence Intervals

We conduct a simulation study to investigate the coverage of the 95% confidence intervals computed with the delta method and the bootstrap method. Both methods are briefly explained in the following.

Using the delta method, the variance of an estimated quantile $q_{d,p}$ (see Equation (9)) can be calculated as follows [14]:

$$\text{Var}(q_{\hat{d},p}) \approx \nabla q_{d,p}^T V \nabla q_{d,p}, \tag{A1}$$

where $V$ is the variance-covariance matrix of the parameter estimations $(\hat{\tilde{\mu}}, \hat{\sigma}_0, \hat{\xi}, \hat{\theta}, \hat{\eta})$ and $\nabla q_{d,p}$ is the gradient

$$\nabla q_{d,p} = \left( \frac{\partial q_{d,p}}{\partial \tilde{\mu}}, \frac{\partial q_{d,p}}{\partial \sigma_0}, \frac{\partial q_{d,p}}{\partial \xi}, \frac{\partial q_{d,p}}{\partial \theta}, \frac{\partial q_{d,p}}{\partial \eta} \right)^T. \tag{A2}$$

Assuming that the maximum likelihood estimator $q_{\hat{d},p}$ follows a normal distribution, the confidence intervals can be calculated from the variance $\text{Var}(q_{\hat{d},p})$. However, this assumption may be poor. To take into account the dependence of the maxima of different durations and stations in the estimation of uncertainties, the delta method can be adjusted accordingly [24]. In this case, however, we did not apply this adjustment, since we verified the coverage only for confidence intervals estimated from independent maxima.

A common and simplistic method for estimating confidence intervals without the assumption of normality is bootstrapping. Here we apply the ordinary non-parametric bootstrap percentile method. Therefore, firstly a sample is created by drawing from the data with replacement, and then the model parameters are estimated. From those, we calculate the resulting return levels. By 1000 repetitions of this process, we obtain a distribution of return levels. We estimate the lower and upper bound of the confidence interval from the empirical 0.025 and 0.975 quantiles of the distribution of return levels. When we apply the bootstrap method to the observations in the study area, we sample the years by randomly drawing with replacement. Therefore all available maxima for a given year are

used collectively in the bootstrap sample. We expect that in this way the dependency structure of the observations is taken into account.

To analyze the coverage of both described methods, we proceed as follows: We draw random values from a d-GEV distribution with the parameters $\tilde{\mu} = 3, \sigma_0 = 5, \xi = 0.06, \theta = 0.05, \eta = 0.7$ to get a sample of size $n$. The parameters are chosen to be comparable to those estimated for individual stations in the research area. However, in this sample the maxima are independent of each other, contrary to what we expect for the observations. For this sample, the 95%-confidence intervals are estimated for a given quantile $q_{d,p}$ using both methods. Then it is tested whether the actual value for $q_{d,p}$ is included in the estimated confidence intervals. This process is repeated 1000 times. Finally, the coverage of the confidence interval is estimated from the relative frequency of how often the actual value was within the confidence interval. The results are presented in Figure A1. The coverage for the delta method intervals varies with duration and probability and deviates strongly from 95%. The bootstrapped confidence intervals, on the other hand, show a reasonable behavior.

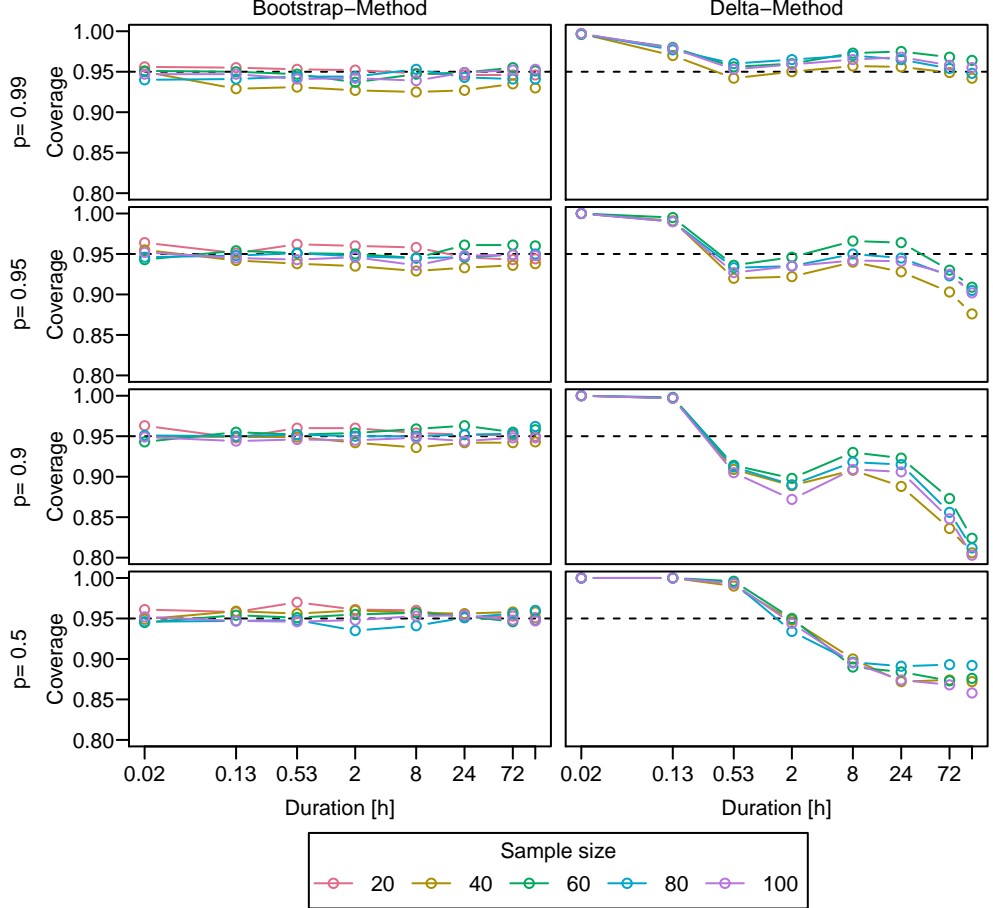

**Figure A1.** Coverage of Confidence Intervals, obtained by the bootstrap method (left column) and the delta method (right column). The coverage was calculated by re-sampling from a known d-GEV distribution 1000 times. Different colors indicate different sample sizes, which correspond to the length of the time series in years in this context. Different rows represent different non-exceedance probabilities $p$, whereby the confidence intervals were examined for the corresponding quantile estimates.

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
