# Peer review of "Estimating IDF Curves Consistently over Durations with Spatial Covariates"

_water, doi:10.3390/w12113119_

Round 1

Reviewer 1 Report

The manuscript describes a method to estimate the intensity-duration-frequency curve for ungauged stations considering spatial covariates. The manuscript is well-written and structured. The figures are informative and discussed. I have several comments:

  1. Avoid using acronyms in the abstract. Better to fully spell GEV, IDF in the abstract.
  2. Line 112, what is “iid”?
  3. Equation 4, is Sigma_Zero defined?
  4. Figure 8 shows that the widths of boxes for duration >16 h are much wider than others. It‘s possible to plot them using the same width to avoid any confusion.
  5. Section 4 “Discussion” includes some statements to summarize the manuscript. I would suggest moving relevant sentences to a new section “Summary”.

Author Response

Dear Reviewer,

We appreciate your constructive comments, which we have now addressed in the manuscript. We have marked major changes in the manuscript in red. Purely language changes are indicated in orange. In the following we will respond to each of your points.

Point 1: Avoid using acronyms in the abstract. Better to fully spell GEV, IDF in the abstract.

Response 1: We have implemented this accordingly, thank you for this advice.

Point 2: Line 112, what is “iid”?

Response 2: We now omit this abbreviation and have replaced it with 'independent and identically distributed' (Line 129).

Point 3: Equation 4, is Sigma_Zero defined?

Response 3: Thanks for pointing out that the definition was missing, we have changed the paragraph accordingly. Sigma_Zero is a non-negative parameter, that can be interpreted as a scale offset, since it indicates the scale parameter of the GEV distribution for the duration d=1-Theta.

Point 4: Figure 8 shows that the widths of boxes for duration >16 h are much wider than others. It‘s possible to plot them using the same width to avoid any confusion.

Response 4: We expanded the explanation in the caption: The width is further
useful information, it represents the amount of data used.
For the corresponding station 14 years of data are available for less than 24 hours, while 87 years of daily measurements are available, which can then be accumulated to multiples of 24 hours.
It would be possible to use the same width for all boxes, however, we feel that this is important information to provide. 

Point 5: Section 4 “Discussion” includes some statements to summarize the manuscript. I would suggest moving relevant sentences to a new section “Summary”.

Response 5: We have implemented this according to your advice.

Sincerely,
Jana Ulrich (on behalf of all authors)

Reviewer 2 Report

The manuscript entitled: Estimating IDF curves consistently over durations with spatial covariates (WATER 962594) proposes a method for obtaining reliable estimates of the distribution of extreme precipitation. This estimate appears to be successful with several important limitations, including that relating to short-term rainfall. As you know, events with a short duration are resolved with different methodologies, and therefore not comparable to the parameters used for those with a duration greater than 1 hour. I would suggest you delve into the various studies that have been successful in this specific estimate. They also work with a smaller number of parameters than those proposed (for example, the classics Bell, 1969; Koutsoyiannis, 2006 and others more recent) and allow a much simpler verification. Your approach involves the integration of spatial covariates for the parameters of a duration-dependent GEV (d-GEV) to model extreme precipitation, both in space and over a time interval. However, the reader is not aware of the context (the basin of the River Wupper in western Germany) nor the choice of this (rather complex) "mathematical method", including the assumptions - bias - about the short-term phenomenon. Your results seem not to be quite brilliant, as you yourself indicated, both because of the "strong" assumptions of the model "and because experimentally - for example - rare events are well highlighted, but not short-lived events. Not knowing this approach, you I would suggest comparing this model with other more well-known ones, testing them in the same basin. In this way you will justify the use of this application. Furthermore, I would propose to include in the reference articles that can support this analysis with "case studies" rather than with methodological ones. Journal, issue and pages would need to be added to some of the articles cited. Therefore, I would propose to improve the manuscript (see also the enclosed file), and therefore request major revisions.

Author Response

Dear Reviewer,

We appreciate your constructive comments, which we have now addressed in the manuscript. We have marked major changes in the manuscript in red. Purely language changes are indicated in orange. In the following we will respond to each of your points.

Point 1: As you know, events with a short duration are resolved with different methodologies, and therefore not comparable to the parameters used for those with a duration greater than 1 hour. I would suggest you delve into the various studies that have been successful in this specific estimate. 

Response 1: The aim of this article is to model exceedance probabilities
(return-levels) across a wide range of durations simultaneously,
including also relevant short durations. Advantages of having just one
model to describe exceedance probabilities for all (relevant)
durations are:
a) A consistent approach in the sense, that quantiles will never
cross, even if extrapolated for lager/smaller durations. We consider
this as a very important constraint in modeling IDF relations.
b) Estimation of exceedance probabilities for one duration profit
from data available at neighboring durations due to the assumption of
smooth variations across duration.
We pursue to exploit these advantages here. We furthermore discuss to
what extend a potential lack of flexibility can be a disadvantage for
particular durations. This seems to be the case for durations less
than 1 hour, as we discuss. We propose to take on a more
flexible model in a follow-up study but retaining the idea of modeling all
durations simultaneously. We revised the introduction and summary to highlight
this aim even more. 

Point 2: They also work with a smaller number of parameters than those proposed (for example, the classics Bell, 1969; Koutsoyiannis, 2006 and others more recent)

Response 2: The approach we use is based on the classical approach for small durations. It has been proposed by Koutsoyiannis et al. [1] to generate IDF curves in the range of 12 min to 24 h. However, this approach is often used for a wider range of durations. We regret that this was not sufficiently clear from chapter 2.2. We have now extended our explanation of the model (line 138 ff.).
However, this approach is often used for a wider range of durations with the aim to transfer knowledge from the long durations to the short durations, where less data is available.
Hence, we decided to model this wide range of durations and subsequently analyze the model performance for the entire range.
The results of our verification show, of course, as you have correctly pointed out, certain limitations in the range of small durations. However, it was the goal of our study to highlight these limitations in order to improve the model in the future accordingly.

[1] Koutsoyiannis, D.; Kozonis, D.; Manetas, A. A mathematical framework for studying rainfall intensity-duration-frequency relationships. J. Hydrol. 1998, 206, 118–135. doi:10.1016/S0022-1694(98)00097-3.

Point 3: and allow a much simpler verification

Response 3: We investigate the performance of the model using well known methods
from forecast verification, i.e. the quantile verification score (QVS)
[2] which is a proper score [3] designed
to verify particular (extreme) quantiles. We use the skill score
derived from the QVS with a reference which is NOT based on the
assumption of smooth variation across durations but is completely
flexible and thus an ideal reference to challenge our duration
dependent approach here. As this reference is also the approach used as a basis
for official return level estimates from the German Meteorological
Service and others, we consider this as an adequate reference. The
skill score is then used in a
cross-validation setting to give an estimate of out-of-sample
performance, the case we typically have in practice: we want
to specify the exceedance probability for future cases.
We perform this method separately for each duration in order to assess the model performance for each duration regime.
Any simpler
verification approaches would likely not suit our high standards of
model validation.

[2] Bentzien, S.; Friederichs, P. Decomposition and graphical portrayal of the quantile score. Q. J. R. Meteorol.
Soc., 2014, 140, 1924–1934. doi:10.1002/qj.2284.

[3] Gneiting, T.; Raftery, E.A. Strictly Proper Scoring Rules, Prediction, and Estimation, J. Am. Stat. Assoc., 2007, 102:477, 359-378, doi:10.1198/016214506000001437

Point 4: Your approach involves the integration of spatial covariates for the parameters of a duration-dependent GEV (d-GEV) to model extreme precipitation, both in space and over a time interval. However, the reader is not aware of the context (the basin of the River Wupper in western Germany) 

Response 4: The Wupper Catchment has a relatively small area with a sufficient amount of gauge stations, especially with higher measuring frequency (hourly/ every minute). In addition, the stations located there show a great variability in elevation and topographic features.
We agree that this extended information about the catchment is useful additional information for the reader and provide it according to your advice in chapter 2.1 (line 107 ff.). 

Point 5: nor the choice of this (rather complex) "mathematical method", including the assumptions - bias - about the short-term phenomenon.

Response 5: We regret not having clearly expressed our aim of pooling information in order to be able to provide reliable estimates even at stations and in duration ranges for which few measurements are available. We hope that this aim is now more strongly emphasized after revision of the introduction and the methods section. 

Point 6: Your results seem not to be quite brilliant, as you yourself indicated, both because of the "strong" assumptions of the model "and because experimentally - for example - rare events are well highlighted, but not short-lived events.

Response 6: The circumstance that there is much less data for short durations than for long durations is of course a problem. However, this problem is present for all IDF methods. It also complicates the verification of models in the range of small durations.
Therefore we are interested in modeling different durations simultaneously, as this allows more reliable estimation for the short durations.
However, as you correctly pointed out, we were interested in showing both the advantages as well as the limitations of the used approach.

Point 7: Not knowing this approach, you I would suggest comparing this model with other more well-known ones, testing them in the same basin. In this way you will justify the use of this application.

Response 7: As a reference, we intentionally avoid using another model based on
the assumption of a smooth relationship between durations, but rather
use a model which estimates extreme value distributions for
individual durations. We consider this as an adequate challenge for
our approach as the reference is completely flexible.
Although a comparison with other IDF models would of course be interesting and provide additional information, we feel that such a comparison is beyond the scope of this study. 

Point 8: Furthermore, I would propose to include in the reference articles that can support this analysis with "case studies" rather than with methodological ones.

Response 8: Regarding your comment, we would like to point you to the following case studies, which use a similar approach to ours:

Lehmann, E.; Phatak, A.; Soltyk, S.; Chia, J.; Lau, R.; Palmer, M. Bayesian hierarchical modelling of rainfall
extremes. 20th International Congress on Modelling and Simulation, Adelaide, Australia; Piantadosi, J
and Anderssen, RS and Boland, J., Ed., 2013, pp. 2806–2812.

Blanchet, J.; Ceresetti, D.; Molinié, G.; Creutin, J.D. A regional GEV scale-invariant framework for
Intensity–Duration–Frequency analysis. J. Hydrol. 2016, 540, 82–95. doi:10.1016/j.jhydrol.2016.06.007.

Van de Vyver, H. A multiscaling-based intensity–duration–frequency model for extreme precipitation.
Hydrol. Process. 2018, 32, 1635–1647. doi:10.1002/hyp.11516.

Fischer, M.; Rust, H.; Ulbrich, U. A spatial and seasonal climatology of extreme precipitation return-levels:
A case study. Spat. Stat. 2019, 34. doi:10.1016/j.spasta.2017.11.007.

Point 9: Journal, issue and pages would need to be added to some of the articles cited.

Response 9: We regret making this mistake. We have corrected all references
according to your advice. However, it is a feature of the Journal's
Latex Template to print certain information and leave out other (also
given in the bibtex-File). Information, such as month or number is not
included by the template. According to the specifications we have now
added the doi for each reference.

Sincerely,
Jana Ulrich (on behalf of all authors)

Reviewer 3 Report

is an interesting and well-documented study. however, it should be supplemented with some references in the introduction and refer to the analyzes, on the topic of the article, made globally. Also, please better highlight the conclusions by creating a separate chapter.

Author Response

Dear Reviewer,

We appreciate your constructive comments, which we have now addressed in the manuscript. We have marked major changes in the manuscript in red. Purely language changes are indicated in orange. In the following we will respond to each of your points.

Point 1: should be supplemented with some references in the introduction and refer to the analyzes, on the topic of the article, made globally

Response 1: Thank you for this advice, in the introduction we now describe in more detail also other approaches that are used/tested in other countries. 

Point 2: please better highlight the conclusions by creating a separate chapter

Response 2: We have implemented this remark by adding an extra chapter for the summary.

Sincerely,
Jana Ulrich (on behalf of all authors)

Round 2

Reviewer 2 Report

I am grateful to the Authors for having satisfied most of my requests. In my opinion it allowed a significant improvement of the manuscript. Therefore, it is suitable for publication in the present form.

Author Response

The authors thank the reviewer for the constructive comments which have contributed to the improvement of the manuscript.